# Effect of the Heterovalent $Sc^{3+}$ and $Nb^{5+}$ Doping on Photoelectrochemical Behavior of Anatase $TiO_2$

Elena S. Siliavka [1], Aida V. Rudakova [1], Tair V. Bakiev [1], Anna A. Murashkina [1], Petr D. Murzin [1], Galina V. Kataeva [2], Alexei V. Emeline [1] and Detlef W. Bahnemann [1,*]

[1]   Laboratory "Photoactive Nanocomposite Materials", Saint Petersburg State University, 199034 Saint-Petersburg, Russia; lenasil1@mail.ru (E.S.S.); aida.rudakova@spbu.ru (A.V.R.); tairbakiev@gmail.com (T.V.B.); murashkinaaa@mail.ru (A.A.M.); murzinpetrff@gmail.com (P.D.M.); alexei.emeline@spbu.ru (A.V.E.)

[2]   Department of General and Applied Physics, Moscow State University of Civil Engineering, 129337 Moscow, Russia; galvk@mail.ru

[*]   Correspondence: bahnemann@iftc.uni-hannover.de

**Abstract:** In this study, we explored the effect of either Nb or Sc doping at a concentration range of 0.0–1.0 at.% on the physical–chemical and photoelectrochemical behavior of $TiO_2$ anatase electrodes. This behavior was characterized by work function, flat band potential, donor density, spectral dependence of photocurrent and stationary photocurrent measurements. All experimental results are interpreted in terms of the formation of the shallow delocalized polaron states in the case of Nb doping and deep acceptor states induced by Sc doping on $TiO_2$ anatase.

**Keywords:** photoelectrochemistry; metal doping; heterovalent doping; work function; intrinsic defects; titanium dioxide

## 1. Introduction

The heterovalent doping of metal oxide photocatalysts with metal cations of the corresponding type is currently considered a conventional strategy to alter their electronic behavior and, therefore, to change their photoactivity and photosensitivity toward visible light [1–4]. It has been established that metal cation doping changes the electronic and optical properties of photocatalysts either through the formation of corresponding dopant impurity defect states or through the stabilization of the intrinsic defects. According to the electronic theory of catalysis [5–7], the presence of new energy levels of such defects leads to an alteration of the catalyst activity in chemical and photochemical reactions due to a shift of the Fermi level position.

The effect of metal cation doping on the photocatalytic behavior of titanium dioxide depends on the chemical nature of the metal cations, their ionic size and charge, and the dopant concentration, as demonstrated by numerous experimental and theoretical studies [1,8–12]. Quite often transition metals are used as heterovalent dopants due to their electronic structure [13]. It is important to emphasize that heterovalent doping always results in the formation of charge-compensating intrinsic defects, the type of which is determined by the difference in charge between dopant state $M^{x+}$ and host cation $Ti^{4+}$ and the concentration of the dopant. A theoretical model describing the correlation between photocatalytic activity, doping ratio, and particle size was proposed by Bloh et al. [14]. According to this approach based on empirical data, the optimal doping ratio to achieve the maximal photoactivity of a particular powdered photocatalyst corresponds to a sufficient number of dopant atoms, i.e., having at least one dopant per particle and not too many dopant atoms per particle to avoid a formation of recombination centers.

Theoretical modeling [12,15,16] infers that the substitutional heterovalent dopants in anatase $TiO_2$ induce the formation of electron and hole states, depending on the type of

dopant, respectively, which can be described in terms of polaron states. DFT calculations demonstrate that the hole polarons in anatase are strongly localized and form deep energy states within the band gap. Thus, these polaron states can be described as deep traps. In turn, the localization of electron-based polarons is rather weak, and polaron states can be extended to several lattice constants. The energy depth of the electron traps is estimated to be less than 0.1 eV with stabilization energies of polaron formation of 0–0.2 eV [17]. It means that electron-based polarons can be described as very shallow electron traps. Consequently, acceptor dopants, such as Al, Ga, In, Sc, Y, induce the formation of deep hole polaron states, while donor dopants, such as Nb, Sb, Ta, V, give rise to the formation of partly delocalized shallow electronic polaron states.

Such behavior of intrinsic defects in doped anatase is completely opposite to that in doped rutile due to the deep localization of electron polarons in rutile [12,16,18,19]. Thus, as it was shown by quantum chemistry modeling, the hole polaron bound at $In^{3+}$ in anatase is about as deep as the states induced by indium ion in rutile. At the same time, the shallow states in anatase and the deep states in rutile are formed for the Nb donor doping [16].

The electronic properties of Nb-doped anatase, where niobium is substituted with a $Ti^{4+}$ site, have been evaluated by measurements of electrical conductivity [20–24]. It was found that transparent anatase $Ti_{1-x}Nb_xO_2$ films with $x \geq 0.01$ exhibit metallic behavior, and, therefore, can be used as a transparent conductive oxide. On the other hand, the transformation of an n-type semiconductor into a degenerate semiconductor can decrease the fraction of light absorbed by titania due to the Burstein–Moss effect, which leads to the blue shift of the optical band gap.

Photocatalytic activity of Nb-doped titanium dioxide (anatase) with Nb content of 0.0–20.0 at.% in the reaction of photodegradation of organic dyes (methylene blue and methyl orange) has been explored for the samples in forms of films and powders prepared by different synthesis methods [20,25–27]. The samples with low doping levels (<1 at.%) were the most effective under UV and visible irradiation, and excessive doping reduced the photocatalytic activity in all cases.

In addition, Nb-doped anatase $TiO_2$ samples demonstrate superior performance in photocatalytic $CO_2$ reduction [28,29] and photocatalytic hydrogen generation [30] under simulated solar illumination. In the latter case, the main $CO_2$ reduction products were methanol (up to 1.00 $\mu mol \cdot g^{-1} \cdot h^{-1}$) for Nb-doped $TiO_2$ powder with an optimal dopant concentration of 2.5% [28] and acetaldehyde (over 500 $\mu mol \cdot g^{-1} \cdot h^{-1}$) for $Ti_{0.95}Nb_{0.05}O_2$ nanotube array [29]. It was suggested that the substitutional $Nb^{5+}$ doping, replacing $Ti^{4+}$, leads to the formation of acidic centers with a positive charge and simultaneously creates $Ti^{3+}$ defects, which enhance both the adsorption and activation of $CO_2$ on the Nb-doped $TiO_2$ surface [29].

The study of the photoelectrochemical behavior of degenerate Nb-doped anatase $TiO_2$ ($Ti_{1-x}Nb_xO_2$: x = 0, 0.01, 0.03, 0.06, 0.1) electrodes prepared by pulsed laser deposition on $LaAlO_3$ and $SrTiO_3$ supports revealed that an increase in Nb concentration causes a significant decay of titania photoactivity [30]. Such observation was explained by a blue shift of the spectral limit of photoactivity and the increase in the efficiency of charge carrier recombination. In contrast to these data, the authors of another report [31] found that the Nb-doped (0–10 at.%) anatase $TiO_2$ films deposited on an ITO glass substrate by spray drying, exhibited the enhanced photoelectrochemical performance under UV irradiation, and the optimum Nb doping concentration is 0.1 at.%.

The Nb-doped anatase $TiO_2$ nanoparticles with Nb contents of 2.5, 5.0, 7.5 and 10.0 mol.% were successfully applied as the photoanode material in dye-sensitized solar cells (DSSCs) [24]. An overall 7.8% energy-conversion efficiency was obtained for a DSSC based on 5.0 mol.% Nb-doped $TiO_2$, which resulted in an improvement of 18.2% relative to that of the undoped $TiO_2$ in DSSC. The improvement was ascribed to the enhanced electron injection and transfer efficiency caused by the positive shift in the flat band potential ($V_{fb}$) and by increased powder conductivity, which was verified by powder resistance and EIS analyses.

In contrast to the Nb-doped anatase $TiO_2$, much less research on the optical and electronic properties of Sc-$TiO_2$ (anatase) can be found. The presented theoretical [32,33] and experimental [34–36] studies show some negligible broadening of the optical band gap for Sc-$TiO_2$.

Only a few studies on the photocatalytic properties of scandium-doped anatase $TiO_2$ have been reported. The efficiency of the $Sc_xTi_{(1−x)}O_2$ (x: 0, 0.005, 0.02 and 0.05 at.%) photocatalyst was verified for the degradation of diclofenac potassium solution under UV light. It was shown that a 5.0 at.% of Sc improves the material's photoactivity [35]. From the experimental data presented in two other studies [37,38], it follows that a combination of the Sc-doped $TiO_2$ semiconductor with electron-donating dopants (either carbon or Ag nanoparticles) increases the activity in the photocatalytic degradation reaction of organic dyes (Acid orange 7, Rhodamine B).

In an extensive study of the effect of 35 dopants on the behavior of Grätzel cells [39], it was noted that 2 at.% of scandium dopant significantly attenuates the light conversion efficiency. The effect of scandium doping of DSSC photoanode made of mesoporous titanium (anatase) was also studied in detail in [36]. The electronic properties of the Sc-doped $TiO_2$ as a function of Sc doping were investigated by the measure of the flat band potential, band gap, and deep-level distribution. In the range 0.0–1.0 at.% of Sc (0, 0.1, 0.2, 0.3, 0.5, 1.0 at.%), the flat band energy changes from −4.15 to −4.07 eV. The presence of Sc cations heavily modifies the cathodoluminescence spectrum of anatase, even at the lowest concentration. Several DSSCs with photoanodes at different Sc doping were tested both under a solar simulator and in the dark. The maximum efficiency of 9.6% was found at 0.2 at.% of Sc in anatase, which is 6.7% higher with respect to the DSSCs with undoped anatase. These observations do not generally contradict the data given in the reference [39] and suggest a tendency for the Sc-doped $TiO_2$-based photoanode efficiency to decrease with increasing the scandium content.

In this work, we continue our studies on the effect of heterovalent doping of titanium dioxide on the defect distribution and photocatalytic and photoelectrochemical behavior of titania [40–44]. Here, we present a comparative study of the effect of doping with trivalent $Sc^{3+}$ cations and pentavalent $Nb^{5+}$ cations on the photoelectrochemical activity of titanium dioxide (anatase) in the dopant concentration range of 0.0 ÷ 1.0 at.% with a small step of 0.2 at.%. Scandium and niobium cations were chosen for heterovalent doping of $TiO_2$ for several reasons. On the one hand, replacing the $Ti^{4+}$ cations in the $TiO_2$ matrix with $Sc^{3+}$ and $Nb^{5+}$ cations brings an excess of the negative and positive charge, respectively. On the other hand, the lattice distortion arises from the difference in the radii of the host and dopant cations (0.605 Å for $Ti^{4+}$ and 0.640 Å for $Nb^{5+}$ or 0.745 Å for $Sc^{3+}$ [45]), as has been demonstrated mainly in the case of Sc-doped $TiO_2$ [34,35]. Both doping effects can induce the formation of intrinsic defect states. It is wise to note that no energy levels corresponding to the electronic states of $Sc^{3+}$ and $Nb^{5+}$ are located within the band gap of $TiO_2$. Thus, the purpose of this study is to explore the effect of the possible redistribution of intrinsic defects depending on the type and concentration of the dopant, either $Nb^{5+}$ or $Sc^{3+}$ on the photoelectrochemical behavior of $TiO_2$.

## 2. Results and Discussions

### 2.1. Physical–Chemical Characterization of the Doped Electrodes

SEM images of all prepared electrodes show that FTO substrates are completely covered by uniform layers formed by 20 nm particles with an average thickness of the $TiO_2$ layers about 200–220 nm. Figure S1 demonstrates SEM microphotographs of the surface and cross-section for the 1.0-Sc-$TiO_2$ electrode, as an example.

The XRD data (Figure S2) proves the formation of the anatase phase for all synthesized samples.

Figure 1 demonstrates the dependencies of the electrode work functions on the dopant concentration for the Nb-doped (curve 1) and Sc-doped (curve 2) sets of the samples.

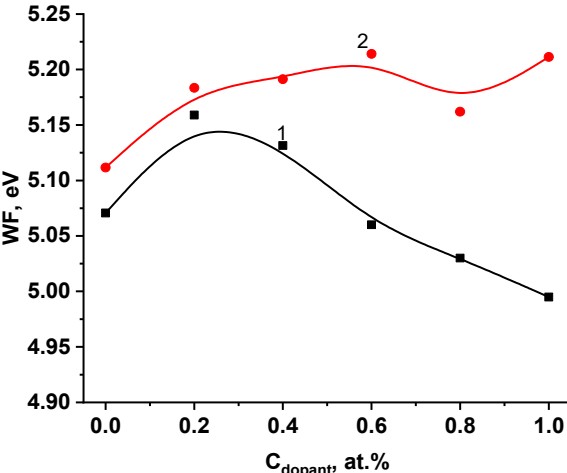

**Figure 1.** Dependencies of the electrode work function for the Nb-doped (1) and Sc-doped (2) sets of the samples on the dopant concentration.

As expected in general, TiO$_2$ doping with Nb results in a decrease in the work function due to the formation of Nb-stabilized shallow polaron states. The higher the Nb concentration, the shallower the electron traps formed, leading to a decrease in the work function. At the same time, doping of TiO$_2$ with Sc leads to an increase in the work function since Sc doping stabilizes deep localized hole polaron states that shift Fermi level position toward the valence band of TiO$_2$.

Figure 2 demonstrates the impedance experimental data presented in the form of the Mott–Schottky plot.

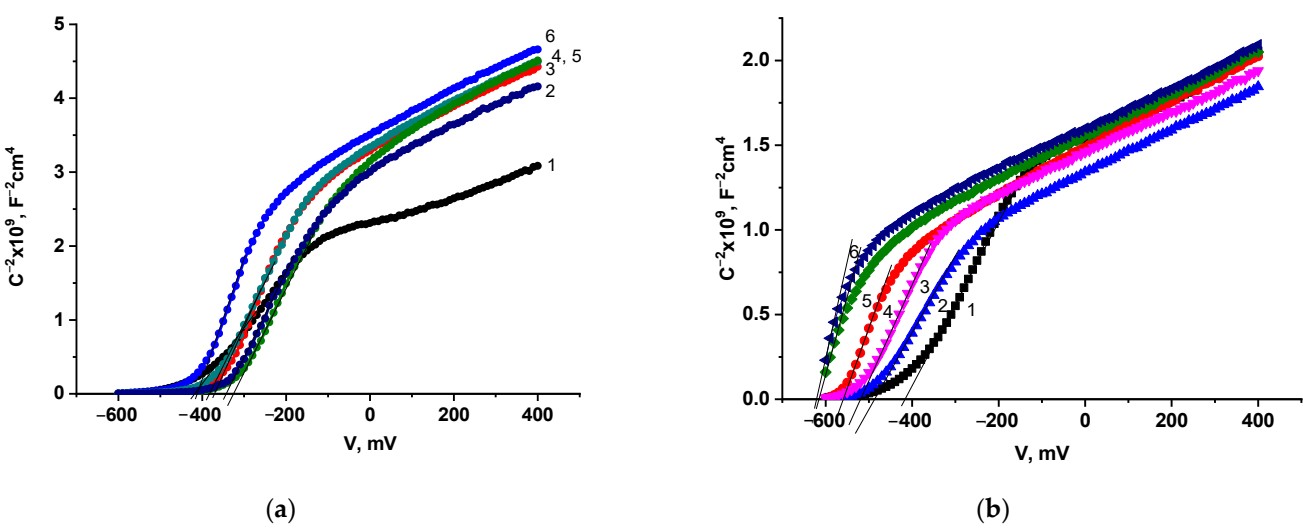

(**a**)                                                      (**b**)

**Figure 2.** Mott–Schottky plot for the sets of Nb-doped (**a**) and Sc-doped (**b**) TiO$_2$ electrodes with the dopant content: 1—0.0 at.%, 2—0.2 at.%, 3—0.4 at.%, 4—0.6 at.%, 5—0.8 at.%, 6—1.0 at.%.

Analysis of the Mott–Schottky dependencies according to Equation (1):

$$C^{-2} = \frac{2}{qA^2\varepsilon N_d}(V + V_{fb}) \tag{1}$$

where $C$ is a capacitance, $A$ is an area of the working surface of electrodes, $\varepsilon$ is a permittivity, N$_d$ is the density of the donor states, $V$ is an applied potential, and $V_{fb}$ is a flat band potential, which allows us to determine how an alteration of the type and concentration of

the dopants, either Nb or Sc, affects the type of major charge carriers, flat band potentials, and density of the donor states.

According to the data presented in Figure 2 (the slopes of the curves), all electrode samples demonstrate n-type conductivity, regardless of the type and concentration of dopants. The values of the slopes ($k$) are proportional to the reciprocal donor concentration ($N_d$) (Equation (1)). Figure 3 demonstrates the dependencies of the donor state densities on the dopant concentrations.

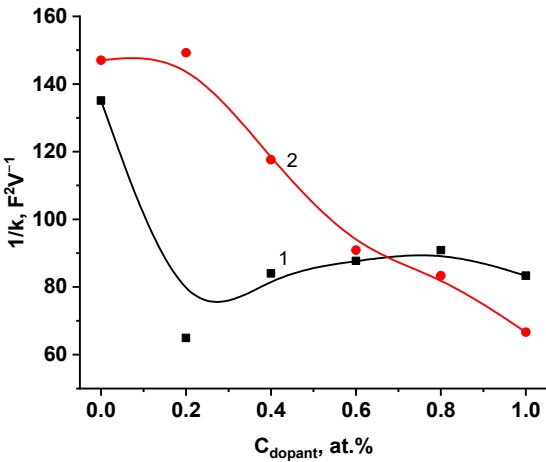

**Figure 3.** Dependencies of the reciprocal slope values, proportional to the density of the donor states, on the dopant concentrations: 1—Nb, 2—Sc.

As evident from the presented dependencies (Figure 3), the density of the donor states for Nb-doped remains almost independent of the dopant concentration, whereas the density of the donor states for Sc-doped $TiO_2$ is gradually decreased with increasing dopant concentration. As mentioned earlier in the Introduction, Nb doping leads to the formation of very shallow polaron states, which can be considered as partly free electrons and, therefore, Nb doping results in an increase in the free electron concentration but not the donor states defects. At the same time, Sc doping leads to the formation of the deep acceptor states within the band gap, which results in the decay of the donor state density with the increase in the dopant concentration. These results correlate with the work function dependencies (Figure 1): an increase in the free electron concentration in the conduction band due to Nb doping leads to a decrease in the work function, whereas a decay of the donor states and formation of the deep acceptor states caused by Sc doping results in stabilization of the work function at higher values compared to the pristine $TiO_2$.

Figure 4 shows the dependencies of the flat band potentials determined from the Mott–Schottky plots (Figure 2) on the dopant concentrations. The increasing number of the shallow polaron states and, as a consequence, the density of the free electrons in the conduction band results in a significant screening of the surface potential and, therefore, in the decay of the flat band potential with the increase in the Nb content. At the same time, the formation of the deep acceptor states caused by Sc doping of $TiO_2$ results in the stabilization of the higher flat band potentials compared to the flat band potential of pristine $TiO_2$.

Thus, the summary of the results of the physical–chemical characterization of the doped electrodes confirms the theoretical predictions that Nb doping of $TiO_2$ anatase leads to the formation of shallow partly delocalized polaron states, whereas Sc doping stimulates the formation of the deep acceptor states. In turn, this alteration in the distribution of the dominating electronic states in $TiO_2$ caused by doping affects the whole electronic subsystem of $TiO_2$ (including its work function values) and results in the significant alteration of the subsurface potential barrier of the electrodes.

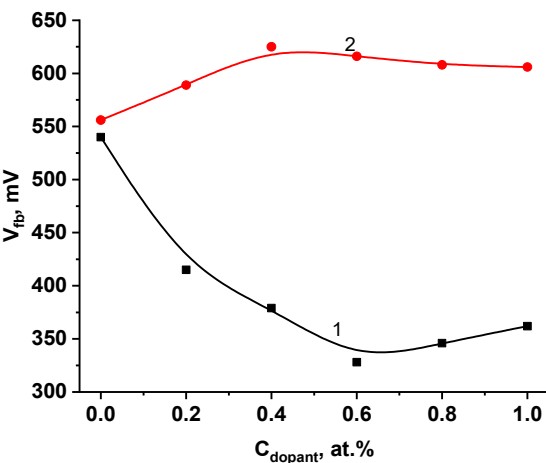

**Figure 4.** Dependencies of the flat band potentials of the electrodes on the dopant concentrations: 1—Nb, 2—Sc.

*2.2. Photoelectrochemical Studies of the Doped Electrodes*

Figure 5 shows the spectral dependences of incident photon-to-current conversion efficiency (PCCE) for nominal pure and Nb and Sc-doped $TiO_2$ electrodes.

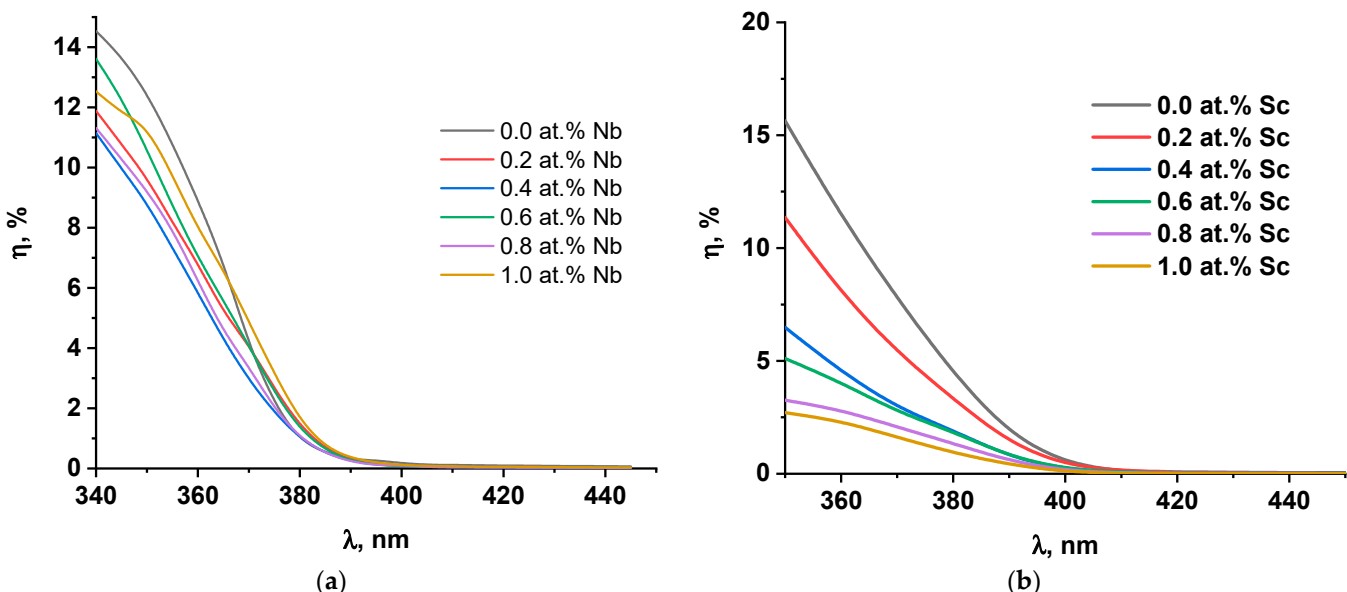

(**a**)   (**b**)

**Figure 5.** Spectral dependencies of PCCE parameter for Nb (**a**) and Sc (**b**)-doped $TiO_2$ electrodes.

As evident from the presented spectral dependencies Nb doping does not much affect the photoelectrochemical efficiency of the corresponding electrodes, whereas Sc doping results in significant decay of photoelectrochemical efficiency of $TiO_2$ electrodes: the higher the Sc dopant concentration, the lower the efficiency. Thus, Sc doping significantly decreases the PCCE parameter for $TiO_2$ electrodes.

Spectral dependencies of PCCE parameters can be used particularly, to determine the band gap values of the doped $TiO_2$ electrodes assuming that PCCE at the edge of the fundamental absorption is proportional to the absorption coefficient. Then, an application of the modified Tauc plot approach let us estimate the band gap values for each electrode (Figure S3). This treatment infers that neither Nb nor Sc doping affects the band gap energy of the doped materials, giving an average estimation of the band gap value $3.18 \pm 0.03$ eV, which is in good agreement with the band gap energy of anatase $TiO_2$ (3.21 eV).

Figure 6 demonstrates the time evolution of photocurrent for Nb and Sc-doped sets of TiO$_2$ electrodes.

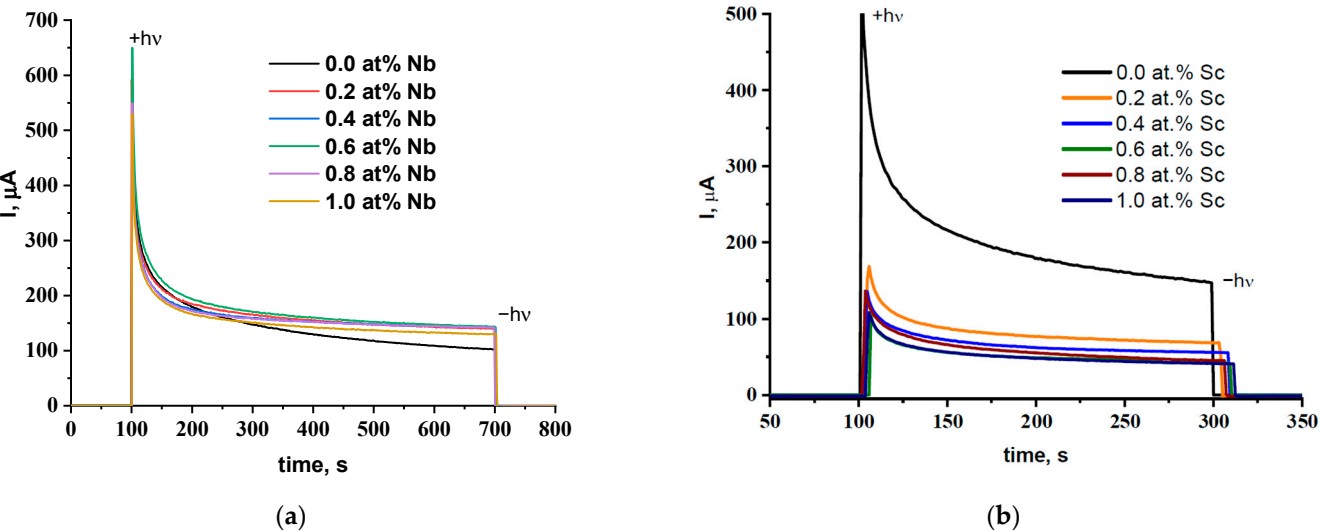

**Figure 6.** Time evolution of photocurrent of Nb (**a**) and Sc (**b**)-doped TiO$_2$ electrodes.

In accordance with spectral dependencies of PCCE, the photocurrent is only slightly changed for Nb-doped TiO$_2$ electrodes and significantly decayed for Sc-doped TiO$_2$ electrodes with increasing the dopant content.

All time dependencies of photocurrent demonstrate maximal values at the initial moment of irradiation, followed by exponential decay approaching the stationary values of the photocurrent. The dependencies of the stationary values of photocurrent and decay time on the dopant content are presented in Figures 7 and 8, respectively.

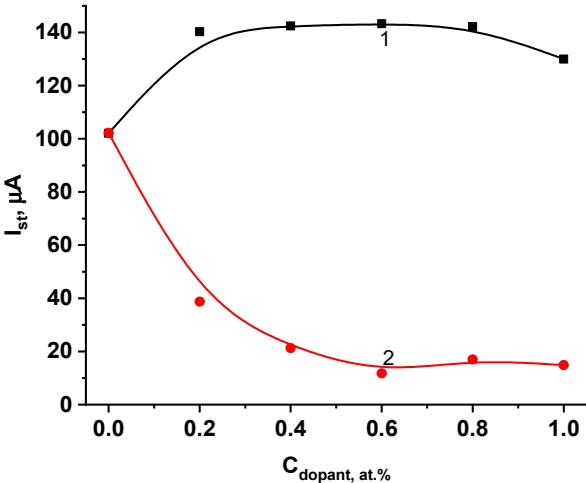

**Figure 7.** Dependencies of the stationary photocurrent values of Nb (1) and Sc (2)-doped TiO$_2$ electrodes on the dopant content.

A slight increase in the stationary photocurrent can be attributed to the decay of recombination efficiency and the resistivity due to the redistribution of the defect states and formation of the shallow polaron states at the bottom of the conduction band caused by Nb doping. In turn, these dominating states result in a shorter decay time of photocurrent since they are not involved significantly in the charge carrier trapping. In opposite, Sc doping leading to the formation of the deep localized acceptor states can be the active charge carrier traps that result in the decrease in the photocurrent and increase in the decay time.

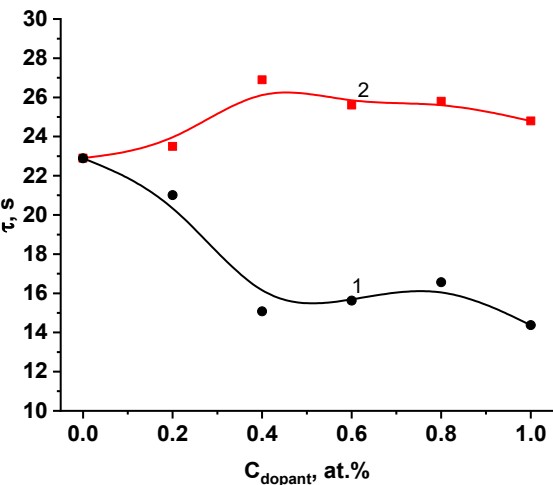

**Figure 8.** Dependencies of the photocurrent decay time on the dopant content: 1—Nb, 2—Sc.

## 3. Materials and Methods

### 3.1. Electrode Preparation

The pristine Sc and Nb-doped $TiO_2$ samples were synthesized by sol–gel method. Sols were prepared using titanium(IV) isopropoxide (TTIP, 97%, Sigma-Aldrich, Darmstadt, Germany), scandium(III) or niobium(V) isopropoxide and isopropanol ($\geq$99.0%, Vekton, Saint-Petersburg, Russia) as titanium and dopant (scandium or niobium) precursors and solvent, respectively. Scandium/niobium isopropoxide solutions were synthesized by dissolving scandium(III) chloride (99.9%, Sigma-Aldrich)/niobium(V) pentachloride (99.9%, Sigma-Aldrich) in isopropanol ($\geq$99.0%, Vekton) adding glacial acetic acid as pH adjusting agent (molar stoichiometry of metal ions to acid was taken as 1:2). Resulting solutions were stirred thoroughly at 40 °C and aging for 48 h. To prepare x-Sc-$TiO_2$ or x-Nb-$TiO_2$ sols, the corresponding volumes of dopant solution were added making the amount of dopant in the final solution corresponding to the required dopant content x.

The dopant concentrations in prepared sols were determined by an inductively coupled plasma atomic emission spectroscopy using a Shimadzu ICPE-9000 ICP emission spectrometer (Shimadzu, Kyoto, Japan).

Polycrystalline x-Sc-$TiO_2$ and x-Nb-$TiO_2$ films, where x = 0.0, 0.2, 0.4, 0.6, 0.8, and 1.0 at.%, were formed by a dip-coating deposition (KSV Nima dip coater, Espoo, Finland) of the corresponding sol solutions on FTO glass substrates (15–25 $\Omega$/cm, Aldrich, Darmstadt, Germany). The substrates were pretreated ultrasonically in isopropanol and annealed at 400 °C.

To achieve the desired thickness of the electrode coatings, the six consequent layers were deposited using a slow withdrawal velocity of 20 mm/min followed by 30 min drying at 100 °C after each dipping cycle. Then, the obtained films were annealed at 500 °C in the air for 4 h (heating rate 60°/h).

### 3.2. Sample Characterization

X-ray diffraction measurements were performed with Bruker D8 Discover high-resolution diffractometer (CuKa X-ray radiation, $20° \leq 2\theta \leq 80°$, scanning speed of 5.0°/min) for the crystal phase determination of all synthesized samples. Reference data were taken from ICSD database. XRD patterns are presented in Figure S2. According to XRD data, both undoped and doped $TiO_2$ films were crystallized in the anatase phase. No formation of any scandium- or niobium-containing phases was observed for all doped samples.

The surface morphology and film thickness of all synthesized films were explored by scanning electron microscopy (Zeiss Supra 40 VP system, Germany).

The dopant concentrations in prepared sols were determined by an inductively coupled plasma atomic emission spectroscopy using a Shimadzu ICPE-9000 ICP emission

spectrometer. The dopant contents in all scandium- and niobium-doped titania samples were confirmed by energy-dispersive X-ray spectroscopy (EDX) and corresponded well to the $Sc^{3+}$ and $Nb^{5+}$ concentrations taken for the synthesis. Dopant contents are summarized in Table 1.

**Table 1.** The dopant content in all scandium (x-Sc-TiO$_2$) and niobium (x-Nb-TiO$_2$)-doped titania samples determined by an energy-dispersive X-ray spectroscopy.

| x-Sc-TiO$_2$ | Sc Content, ат.% | x-Nb-TiO$_2$ | Nb Content, ат.% |
|---|---|---|---|
| 0-Sc-TiO$_2$ | 0.00 | 0-Nb-TiO$_2$ | 0.00 |
| 0.2-Sc-TiO$_2$ | 0.25 | 0.2-Nb-TiO$_2$ | 0.22 |
| 0.4-Sc-TiO$_2$ | 0.45 | 0.4-Nb-TiO$_2$ | 0.38 |
| 0.6-Sc-TiO$_2$ | 0.60 | 0.6-Nb-TiO$_2$ | 0.64 |
| 0.8-Sc-TiO$_2$ | 0.82 | 0.8-Nb-TiO$_2$ | 0.81 |
| 1.0-Sc-TiO$_2$ | 1.07 | 1.0-Nb-TiO$_2$ | 1.08 |

Work function measurements were performed by scanning Kelvin probe system SKP5050 (KP Technology, Caithness, Scotland) using a golden probe electrode (probe area 2 mm$^2$) as a reference. The probe oscillation frequency was 74 Hz, and the backing potential was 7000 mV. Work function values were obtained by averaging 50 data points for five different spots at each sample.

*3.3. Photoelectrochemical Measurements*

The PEC performance of the synthesized electrodes was tested using a three-electrode photo-electrochemical cell. An FTO glass with a deposited sample was employed as the working electrode, and a Pt wire and an Ag/AgCl (0.222 V vs. NHE potential) were used as the counter electrode and reference electrode, respectively.

The chronoamperometry (I-t) curves, cyclic voltammogram (j-V) plots, and Mott–Schottky plots were measured in 0.2 M K$_2$SO$_4$ electrolyte (pH 6.98) using an Elins Pi-50Pro-3 potentiostat/galvanostat and an Elins Z-1500J impedancemeter (LLC "Elins", Moscow, Russia). Current–voltage dependences (I-V curves) were recorded in the voltage region from –1.0 V to +1.0 V using a scan rate of 10 mV/s.

A 150 W Xe lamp (LOMO, Saint-Petersburg, Russia) was used as a light source. The spectral dependences of photocurrent were carried out with the setup consisting of a 150 W Xe lamp with a water filter and MDR-12 monochromator (LOMO, Saint-Petersburg, Russia) and a set of color filters (Vavilov State Optical Institute, Saint-Petersburg, Russia). The spectral resolution of the wavelength dependence measurements was about $\Delta\lambda = \pm 2.5$ nm.

## 4. Conclusions

Theoretical consideration [17,18] predicts that Nb doping of TiO$_2$ anatase can result in the formation of very shallow significantly delocalized electron polaron states at the bottom of the conduction band. Experimental results presented in this paper are in good agreement with this statement. Indeed, the formation of shallow polaron states must lead to a lower recombination efficiency and a higher density of the free electrons in the conduction band. The consequences of these changes will be lower work function and flat band potential, higher conductivity and photoelectrochemical efficiency in complete agreement with the observed experimental results. In contrast to Nb doping, according to the theoretical predictions, Sc doping of TiO$_2$ results in the formation of deep strongly localized states within the band gap [15] that, in turn, must result in higher recombination efficiency, larger work function and flat band potential, and lower photoelectrochemical efficiency, which completely corresponds to the observed effects of Sc doping on electronic and photoelectrochemical behavior of TiO$_2$.

**Supplementary Materials:** The following supporting information can be downloaded at: https://www.mdpi.com/article/10.3390/catal14010076/s1, Figure S1: SEM images of surface and cross-section for 1.0-Sc-TiO$_2$ with 1.0 at.% Sc. Mott–Schottky plots for x-Sc-TiO$_2$/FTO and x-Nb-TiO$_2$/FTO electrodes; Figure S2: X-ray patterns for x-Sc-TiO$_2$ (a) and x-Nb-TiO$_2$ (b) with different dopant concentrations: (1) 0 at.%, (2) 0.2 at.%, (3) 0.4 at.%, (4) 0.6 at.%, (5) 0.8 at.%, (6) 1.0 at.%; Figure S3: spectral dependencies of the modified Tauc plots for Nb (a) and Sc (b)-doped TiO$_2$ electrodes.

**Author Contributions:** Conceptualization, A.V.E. and D.W.B.; methodology, A.V.R.; software, T.V.B.; validation, A.V.E., A.V.R. and A.A.M.; formal analysis, G.V.K., A.A.M. and T.V.B.; investigation, T.V.B., P.D.M. and A.A.M.; resources, E.S.S. and A.V.R.; writing—original draft preparation, A.A.M., A.V.R. and A.V.E.; writing—review and editing, A.V.E. and D.W.B.; visualization, T.V.B., A.A.M. and A.V.R.; supervision, A.V.E.; project administration, A.V.E. and D.W.B.; funding acquisition, A.V.E. and D.W.B. All authors have read and agreed to the published version of the manuscript.

**Funding:** The reported study was funded by the Russian Science Foundation, project number 22-13-00155.

**Data Availability Statement:** Data are contained within the article and Supplementary Materials.

**Acknowledgments:** Experimental studies were performed in the laboratory "Photoactive nanocomposite materials" supported by the Saint-Petersburg State University (ID: 91696387). The authors are thankful to the Resource Center (RC) "Nanophotonics", RC "X-ray Diffraction Studies", RC "Chemical Analysis and Materials", RC "Centre for Physical Methods of Surface Investigation", RC "Centre for Diagnostics of Functional Materials for Medicine, Pharmacology and Nanoelectronics", RC "Nanotechnology" and RC "Geomodel" of the Research Park at the Saint-Petersburg State University for helpful assistance in the preparation and characterization of the samples.

**Conflicts of Interest:** The authors declare no conflict of interest.

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
