# Peer review of "Effect of the Heterovalent Sc3+ and Nb5+ Doping on Photoelectrochemical Behavior of Anatase TiO2"

_catalysts, doi:10.3390/catal14010076_

Round 1

Reviewer 1 Report

Comments and Suggestions for Authors

This paper explored the effect of either Nb or Sc doping at concentration range 0.0 – 1.0 at.% on the physical-chemical and photoelectrochemical behavior of TiO2 anatase electrodes. This behavior was characterized by work function, flat band potential, donor density, spectral dependence of photocurrent and stationary photocurrent measurements. All experimental results are interpreted in terms of formation of the shallow delocalized polaron states in a case of Nb doping and deep acceptor states induced by Sc doping on TiO2 anatase. Overall, the manuscript is well organized. Therefore, I recommend acceptance for publication on Catalysts after the authors have revised their manuscript according to the following comments:

1. Please explain the reasons for the influence of Sc3+ and Nb5+ doping on the photoelectrochemical behavior of rutile TiO2.

2. The manuscript only mentions The XRD data (Figure S2) proves the formation of the anatase phase for all synthesized samples.” without any other description. Please describe the phenomenon and reason for the XRD data results of the rutile TiO2 catalyst doped with Nb or Sc.

3. Please add the data of TiO2 anatase without Nb or Sc doping in manuscript (Figure1, Figures 2…) for comparison.

4. Please add the important results obtained from this experiment in the abstract.

5. The Figures in manuscript are not very clear. Please author provide clear data and Figures.

Comments on the Quality of English Language

This paper explored the effect of either Nb or Sc doping at concentration range 0.0 – 1.0 at.% on the physical-chemical and photoelectrochemical behavior of TiO2 anatase electrodes. This behavior was characterized by work function, flat band potential, donor density, spectral dependence of photocurrent and stationary photocurrent measurements. All experimental results are interpreted in terms of formation of the shallow delocalized polaron states in a case of Nb doping and deep acceptor states induced by Sc doping on TiO2 anatase. Overall, the manuscript is well organized. Therefore, I recommend acceptance for publication on Catalysts after the authors have revised their manuscript according to the following comments:

1. Please explain the reasons for the influence of Sc3+ and Nb5+ doping on the photoelectrochemical behavior of rutile TiO2.

2. The manuscript only mentions The XRD data (Figure S2) proves the formation of the anatase phase for all synthesized samples.” without any other description. Please describe the phenomenon and reason for the XRD data results of the rutile TiO2 catalyst doped with Nb or Sc.

3. Please add the data of TiO2 anatase without Nb or Sc doping in manuscript (Figure1, Figures 2…) for comparison.

4. Please add the important results obtained from this experiment in the abstract.

5. The Figures in manuscript are not very clear. Please author provide clear data and Figures.

Author Response

We are very grateful to reviewers for the attentive and positive consideration of our manuscript and for the useful comments and suggestions.

Reviewer 1

This paper explored the effect of either Nb or Sc doping at concentration range 0.0 – 1.0 at.% on the physical-chemical and photoelectrochemical behavior of TiO2 anatase electrodes. This behavior was characterized by work function, flat band potential, donor density, spectral dependence of photocurrent and stationary photocurrent measurements. All experimental results are interpreted in terms of formation of the shallow delocalized polaron states in a case of Nb doping and deep acceptor states induced by Sc doping on TiO2 anatase. Overall, the manuscript is well organized. Therefore, I recommend acceptance for publication on Catalysts after the authors have revised their manuscript according to the following comments:

We are grateful to Reviewer

 Please explain the reasons for the influence of Sc3+and Nb5+doping on the photoelectrochemical behavior of rutile TiO2.

The main reasons to choose Sc3+ and Nb5+ cations as heterovalent dopants are described in the Introduction (lines 134 – 144). All experimental results presented in the manuscript let us to come to conclusions describing the effect of the heterovalent doping on photoelectrochemical behavior of TiO2 anatase presented in the Conclusion section (lines 388 – 400). Briefly, the heterovalent doping results in redistribution of the intrinsic defect states that in turn, changes the efficiency of recombination and therefore, the photoelectrochemical activity of the doped TiO2.

  1. The manuscript only mentions “The XRD data (Figure S2) proves the formation of the anatase phase for all synthesized samples.” without any other description. Please describe the phenomenon and reason for the XRD data results of the rutile TiO2catalyst doped with Nb or Sc.

It is not clear from the comment what phenomenon we should describe. There is no any specific phenomenon demonstrated by XRD data. XRD data clearly demonstrate only that all pristine and doped samples correspond to the TiO2 anatase phase which is a necessary condition to explore the effect of the heterovalent doping on photoelectrochemical behavior of the material.

  1. Please add the data of TiO2anatase without Nb or Sc doping in manuscript (Figure1, Figures 2…) for comparison.

All figures contain the data points corresponding to the undoped TiO2 (at 0 at.%).

  1. Please add the important results obtained from this experiment in the abstract.

We believe that the Abstract is fully describes the major content and conclusions of the studies presented in the manuscript.

  1. The Figures inmanuscriptare not very clear. Please author provide clear data and Figures.

Unfortunately, it is absolutely unclear from this comment what is not clear in data and Figures to Reviewer.

Reviewer 2 Report

Comments and Suggestions for Authors

Although this is a rather interesting and important article that can be recommended for publication, however some comments/questions should be taken in to account.

1.     Abstract:  All experimental results are interpreted in terms of formation of the shallow delocalized polaron states in a case of Nb doping” and deep acceptor states induced by Sc doping on TiO2 anatase.

Lines 140 – 142. “. It is wise to note that no energy levels corresponding to the electronic states of Sc3+ and Nb5+ locate within the band gap of TiO2”. Don't you think there is an internal contradiction between these two sentences?

2.     Could the authors draw a band diagram and show on it the ground states of the dopants under consideration?

3.     Lines 272-275. “This treatment infers that neither Nb nor Sc doping does not affect the band gap energy of the doped materials giving an average estimation of the band gap  value 3.18 ± 0.03 eV, which is in good agreement with the band gap energy of anatase TiO2 (3.21 eV) “

This is not surprising and absolutely correct, because such small concentrations of impurities cannot change the overall band pattern, but they can visually change the absorption edge due to electronic levels associated with impurities.  See, recent article of the Editors of “Optical Materials”: Brik, M. G., Srivastava, A. M., & Popov, A. I. (2022). A few common misconceptions in the interpretation of experimental spectroscopic data. Optical Materials127, 112276.

4.     Why are vacancies not considered in the work as charge compensators of Sc3+ and Nb5+ ions?

Overall, the manuscript is certainly interesting and should be considered for publication after constructive reflection on the above comments.

Author Response

We are very grateful to reviewers for the attentive and positive consideration of our manuscript and for the useful comments and suggestions.

Reviewer 2

Although this is a rather interesting and important article that can be recommended for publication, however some comments/questions should be taken in to account.

  1. Abstract: “All experimental results are interpreted in terms of formation of the shallow delocalized polaron states in a case of Nb doping” and deep acceptor states induced by Sc doping on TiO2 anatase.

Lines 140 – 142. “. It is wise to note that no energy levels corresponding to the electronic states of Sc3+ and Nb5+ locate within the band gap of TiO2”. Don't you think there is an internal contradiction between these two sentences?

No, there is no contradiction between these two statements. In fact, they correspond well to each other. Briefly, d-orbitals of both Sc3+ and Nb5+ cations locate at higher energy than the bottom of the conduction band of TiO2 and therefore, they do not form localized states within band gap. At the same time, the excess of charge brought by heterovalent dopants requires the formation of the intrinsic defect states of the opposite charge for charge compensation of the dopant cations. For example, the excess of the positive charge of Nb5+ state can be compensated by formation of Ti3+ state. According the theoretical modeling Ti3+ states in anatase can be described as a shallow significantly delocalized polaron state. In turn, excess of the negative charge of Sc3+ cation can be compensated by formation of O-hole state, which in theoretical models can be described as deep strongly localized polaron state.

  1. Could the authors draw a band diagram and show on it the ground states of the dopants under consideration?

Band diagram is presented now in Graphic Abstract.

  1. Lines 272-275. “This treatment infers that neither Nb nor Sc doping does not affect the band gap energy of the doped materials giving an average estimation of the band gap value 3.18 ± 0.03 eV, which is in good agreement with the band gap energy of anatase TiO2 (3.21 eV) “

This is not surprising and absolutely correct, because such small concentrations of impurities cannot change the overall band pattern, but they can visually change the absorption edge due to electronic levels associated with impurities.  See, recent article of the Editors of “Optical Materials”: Brik, M. G., Srivastava, A. M., & Popov, A. I. (2022). A few common misconceptions in the interpretation of experimental spectroscopic data. Optical Materials, 127, 112276.

Yes, we completely agree with all statements and conclusions made in this article. At the same time, we should remind that in our studies the band gap was estimated from the spectral dependencies of the photocurrent, that means that light absorption results in generation of the free charge carriers caused by fundamental absorption.

  1. Why are vacancies not considered in the work as charge compensators of Sc3+ and Nb5+ ions?

Formation of vacancies is certainly not a case for Nb5+ doping as indicating by decrease of the work function, which in turn, indicates the formation of the shallow states. Remarkably, that at higher Nb5+ concentration (> 1.0 at.%) TiO2 anatase can be turned out into degenerated semiconductor state. Thus, theoretical model of the shallow delocalized polaron states corresponds well to the experimental observation. For Sc3+ doping, the formation of anion vacancies as charge compensating defects is quite possible. However, from the charge compensation point of view a formation of a single anion vacancy requires two Sc3+ cations. Thus, most likely that anion vacancies can be formed at higher dopant concentration. In our interpretation we just follow the theoretical model of the deep strongly localized hole polaron state, which does not contradict the experimental results presented in our manuscript. Moreover, our experimental data cannot distinguish between deep hole polaron states and anion vacancies since the accumulation of both types of the defect states should results in the increase of the work function and recombination efficiency.

 Overall, the manuscript is certainly interesting and should be considered for publication after constructive reflection on the above comments.

We are very thankful to Reviewer for the positive interest to our studies and for significant and important questions.

Round 2

Reviewer 2 Report

Comments and Suggestions for Authors

The authors have successfully improved the original version of their manuscript, responding constructively to all the comments/recommendations of the reviewer.  Therefore, the article can be recommended for publication.